# PROTOTYPICAL NETWORKS FOR FEW-SHOT LEARNING

**Jake Snell[1]\*, Kevin Swersky[2]& Richard S. Zemel[1]**
[1]University of Toronto
[2]Twitter

## ABSTRACT

A recent approach to few-shot classification called matching networks has demonstrated the benefits of coupling metric learning with a training procedure that mimics test. This approach relies on an attention scheme that forms a distribution over all points in the support set, scaling poorly with its size. We propose a more streamlined approach, *prototypical networks*, that learns a metric space in which few-shot classification can be performed by computing Euclidean distances to prototype representations of each class, rather than individual points. Our method is competitive with state-of-the-art few-shot classification approaches while being much simpler and more scalable with the size of the support set. We empirically demonstrate the performance of our approach on the Omniglot and *mini*ImageNet datasets. We further demonstrate that a similar idea can be used for zero-shot learning, where each class is described by a set of attributes, and achieve state-of-the-art results on the Caltech UCSD bird dataset.

## 1   INTRODUCTION

One-shot classification (Miller et al., 2000; Lake et al., 2011; Koch, 2015) (and more generally, few-shot classification) is a problem in which a classifier must be adapted to accommodate new classes not seen in training, given only a single ($n$) example(s) of these classes. A classical approach, such as retraining the model on the new data, would severely overfit. While the problem is quite difficult, it has been demonstrated that people have the ability to successfully perform one-shot classification (Lake et al., 2011). Nonparametric models such as nearest neighbors are useful in one-shot classification because they naturally adapt to new data, however this comes at the cost of storing the entire set of examples per class, the "support set".

To overcome this, much progress has been made recently in applying metric learning (Goldberger et al., 2004; Kulis, 2012; Bellet et al., 2013) to one-shot tasks. Most recently, (Vinyals et al., 2016) proposed a metric learning approach that they call matching networks. This approach uses an attention mechanism over a learned embedding of the support set in order to predict class labels for the points to be classified, a.k.a the "query set". It optionally allows the embeddings to be conditioned on other points in the support set ("full context embeddings") or for the embeddings to be fine-tuned at test time. A particularly interesting feature of the matching networks model is that it utilizes sampled mini-batches called "episodes" during training, where each episode is designed to mimic the one-shot task. This makes the training problem more faithful to the test environment. Matching networks however optionally utilize additional components such as an attention-based LSTM to change the embedding based on the support set. This complexity makes implementation more difficult in addition to the aforementioned poor scaling characteristics due to computing attention over the entire support set.

In this paper, we propose a few-shot learning classifier based on a relatively simple idea: there exists an embedding, where points belonging to a class cluster around a single prototype. This inductive bias is a useful one to combat overfitting for one-shot tasks. Our approach also comes with the benefit that it is very simple to implement, and computationally fast. In order to do this, we learn a non-linear mapping of the input into an embedding space using a neural network, and take the class

---

\*Most work done while author was at Twitter.

prototype to be the mean of the support set in the embedding space. Classification is then performed by simply finding the nearest prototype to the embedded query point. We find that this approach yields competitive results with matching networks and other one-shot learning approaches, despite being much simpler.

A related problem is known as zero-shot learning, where instead of being given a small number of examples of a new class at test-time, each class comes with a set of meta-information, often attributes, that give a high level description of that class. The idea then is to learn a mapping from input examples to the high-level attributes of their member class. We adapt the idea of prototypical networks to this setting by learning a secondary embedding of the attribute vector such that the image embeddings and attribute embeddings lie within the same space. In this case, we use the attribute embedding as the class prototype, rather than the class mean.

## 2 RELATED WORK

Neighborhood Components Analysis (NCA) (Goldberger et al., 2004) learns a Mahalanobis distance to maximize K-nearest-neighbour's (KNN) leave-one-out accuracy in the transformed space. A distribution over the neighbors of each data point is computed according to a softmax over the corresponding Mahalanobis distances. This distribution is marginalized to form a distribution over class assignments and the projection matrix is updated via gradient descent to maximize the probability of the true class. (Salakhutdinov & Hinton, 2007) extend NCA by using a neural network to perform the transformation. Our approach is similar in that we optimize a softmax based on distances in the transformed space. Ours differs because it is a softmax over classes, rather than points, computed from Euclidean distances to each class's prototype representation. This is more appropriate for few-shot learning for two reasons: (a) the number of support points can vary by class, and (b) each class has a succinct representation independent of the number of data points, and this representation can optionally be updated in an online manner.

Our approach is similar to the nearest class mean approach of (Mensink et al., 2013) from the metric learning literature, where each class is represented by the mean of its examples, and classification is performed by finding the prototype that is closest to the query point. Their approach was developed to rapidly incorporate new classes into a classifier without retraining, however it relies on a linear embedding and is designed to handle the case where the novel classes come with many examples. In our approach, we utilize neural networks to learn a non-linear embedding of the features and we couple this with episodic training in order to handle the one-shot scenario. Mensink et al. do attempt to extend their approach to perform non-linear classification, but they do this by allowing classes to have multiple prototypes. They find these prototypes in a pre-processing step by using k-means on the input space, and then perform a multi-modal variant of their linear embedding. By contrast, we learn a non-linear embedding in an end-to-end manner with no such pre-processing, producing a non-linear classifier that still only requires one prototype per class.

In matching networks (Vinyals et al., 2016) they propose a meta-learning strategy in which training mimics test by stochastically creating one-shot "episodes". We adopt the same strategy when training our models. They, like us, use neural networks to non-linearly transform data points into a space that is more amenable to classification. However, matching networks make predictions by computing attention weights over each point in the support set. This becomes computationally expensive as the size of the support set grows. Our approach, on the other hand, first summarizes each class in the support set by a prototype and then computes a distribution over classes. Ours thus has flexibility in the way the prototypes are computed and can handle additional support points gracefully by updating prototypes online.

The neural statistician (Edwards & Storkey, 2016) extends the variational autoencoder (Kingma & Welling, 2013) to learn generative models of datasets rather than individual points. One component of the neural statistician is the "statistic network" which summarizes a set of data points into a statistic vector. It does this by encoding each point within a dataset, taking a sample mean, and applying a post-processing network to obtain an approximate posterior over the statistic vector. Edwards & Storkey test their model for one-shot classification on the Omniglot dataset (Lake et al., 2011) by considering each character to be a separate dataset and making predictions based on the class whose approximate posterior over the statistic vector had minimal KL-divergence from the test point. Like the neural statistician, we also produce a summary statistic for each class. However, ours

is a discriminative model which is more appropriate because our primary task, one-shot learning, is also discriminative. Discriminative training has the added benefit of lending our model more flexibility in both the way we compute summary statistics and use them to make predictions at test time.

There are many other approaches to one-shot learning that employ very different techniques from ours. Koch uses siamese networks to predict the probability that two images belong to the same class. Lake et al. devise a hierarchical Bayesian generative model of how a handwritten character is created in order to perform one-shot learning on the Omniglot dataset. Santoro et al. propose memory augmented neural networks (MANN) that reference an external memory in a similar fashion to neural Turing machines (Graves et al., 2014). This allows them to store support examples in an external memory and reference them later when making classification decisions. They also introduce a form of episodic training, similar to that in matching networks.

## 3 PROTOTYPICAL NETWORKS

At prediction time we are given a support set of $N$ labeled examples: $S = \{(x_i, y_i)\}_{i=1}^N = S^1 \cup \ldots \cup S^K$ where $S^k = \{(x, y) \in S \mid y = k\}$. Our method computes a class representation $c_k$, or *prototype*, of each class through an embedding function $f_\theta(x)$ parameterized by learnable parameters $\theta$:

$$c_k = \frac{1}{|S^k|} \sum_{(x,y) \in S^k} f_\theta(x) \tag{1}$$

Given a test point $\tilde{x}$, prototypical networks forms a distribution over classes based on a softmax over the Euclidean distances between its embedding and the prototypes:

$$p(y = k \mid \tilde{x}) = \frac{\exp(-\|f_\theta(\tilde{x}) - c_k\|^2)}{\sum_{k'} \exp(-\|f_\theta(\tilde{x}) - c_k'\|^2)} \tag{2}$$

Learning proceeds by maximizing the log-probability of the true class $\tilde{y}$:

$$\max_\theta \log p(\tilde{y} \mid \tilde{x}) \tag{3}$$

We train in an episodic manner similar to Vinyals et al. (2016) by randomly selecting a subset of classes from the training set, then choosing a subset of examples within each class to act as the support set and the remainder to serve as test points.

### 3.1 PROTOTYPE NORMALIZATION

In episodic training, the support set is randomly chosen from among the training points. In datasets with high variability this can lead to a large variance in the class prototypes, $c$, between episodes. In order to reduce this variability, we found that it can sometimes be beneficial to normalize the prototypes, $\bar{c}_k = \frac{c_k}{\|c_k\|}$ and use $\bar{c}_k$ in place of $c_k$ in Equation (2). This ensures that the prototypes always lie on the unit sphere, although the query points are still allowed to be embedded off of the unit sphere. Normalization has two benefits: the reduction in variance helps to greatly speed up training, while the restriction of the prototypes to the unit sphere confers additional regularization.

### 3.2 PREDICTING THE WEIGHTS OF A LINEAR CLASSIFIER

A simple analysis is useful in gaining insight into the nature of the learned classifier (a similar analysis appears in Mensink et al. (2013)). When we use Euclidean distance to measure the distance between a query point and the class prototypes, then the loss function in (2) is equivalent to a linear classifier with a particular parameterization. To see this, we expand the term within the exponent:

$$-\|f_\theta(\tilde{x}) - c_k\|^2 = -(f_\theta(\tilde{x}) - c_k)^\top (f_\theta(\tilde{x}) - c_k)$$
$$= -f_\theta(\tilde{x})^\top f_\theta(\tilde{x}) + 2c_k^\top f_\theta(\tilde{x})^\top - c_k^\top c_k \tag{4}$$

The first term in Equation (4) is constant with respect to the class $k$, so it does not affect the softmax probabilities. We can write the remaining terms as a linear classifier as follows:

$$2c_k^\top f_\theta(\tilde{x}) - c_k^\top c_k = w_k^\top f_\theta(\tilde{x}) + b_k$$
$$w_k = 2c_k$$
$$b_k = -c_k^\top c_k$$

We can view this through the lens of meta-learning, where the model is predicting the weights and biases of a linear classifier using a simple function of the mean of the embedded support set. By contrast, the predictive function in matching networks is a generalization of a nearest neighbor classifier, rather than a linear classifier.

When using prototype normalization, the biases $b_k$ will all be 1, and the class weights $w_k$ will be restricted to have a norm of 2. In this case, using Euclidean distance becomes proportional to cosine distance.

A natural question is whether it makes sense to use multiple prototypes per class instead of just one. If each support point were to be considered a prototype, then this would be analogous to doing nearest neighbor classification in the embedding space, which would be computationally expensive. On the other hand, if the number of prototypes per class is fixed, then this would require a partitioning scheme. This has been proposed in Mensink et al. (2013) and Rippel et al. (2016), however both methods require a separate partitioning phase that is decoupled from the weight updates, while our approach is simple to learn with ordinary gradient methods. Finally, the equivalence to a linear classifier suggests that this may be sufficient, as all of the required non-linearity can be learned within the embedding function. Indeed, this is the approach that state-of-the-art neural network classification systems currently use, e.g., (Krizhevsky et al., 2012).

## 3.3 Design Choices

There are still a number of design choices that need to be made with this model in order to achieve optimal performance. One such choice is in deciding how many classes we would like the classifier to operate over during each training episode. For example, at test time we might be evaluating on 5-way classification, but at training time we could train each episode with 20-way classification. We found in general that training on a larger number of classes per episode improves performance, even if the number of classes we need to decide between at test-time is fewer.

Another choice involves the possible decoupling of the $n$ in $n$-shot between training and testing. We could train on 1-shot, but test on 5-shot or vice-versa. We found that it is typically better to match the shot at training and testing; that is, when it comes to the shot, to match the training procedure to the test procedure. We demonstrate this empirically in the Experiments section below.

Finally, we need to specify whether to use prototype normalization. We found that normalization generally acts as a regularizer and speeds up training.

## 4 Experiments

For few-shot learning, we performed experiments on Omniglot (Lake et al., 2011), the *mini*Imagenet version of ILSVRC-2012 (Russakovsky et al., 2015) proposed by (Vinyals et al., 2016). For zero-shot learning, we perform experiments on the 2011 version of the Caltech UCSD bird dataset (CUB-200 2011) (Welinder et al., 2010).

### 4.1 OMNIGLOT

Omniglot (Lake et al., 2011) is a dataset of 1623 handwritten characters collected from 50 alphabets. There are 20 examples associated with each character, where each example was drawn by a different human subject. We follow the procedure of (Vinyals et al., 2016) by augmenting the characters with rotations in multiples of 90 degrees and using 1200 characters for training and the remainder for evaluation. Our embedding architecture mirrors that of Matching Nets and is composed of four blocks of a 64 filter $3 \times 3$ convolution, batch normalization (Ioffe & Szegedy, 2015), a ReLU nonlinearity and a $2 \times 2$ max-pooling, resulting in a 64-dimensional output space. The results of our model trained to perform Omniglot classification are shown in Table 1.

We trained prototypical networks using episodes designed for 1-shot learning, i.e., the support sets during training consist of a single input example, and we train using 20-way classification. Our results are as good or better than those reported in matching networks, and to our knowledge represent the state-of-the-art on this dataset using these splits.

| Model | 5-way | | 20-way | |
|---|---|---|---|---|
| | 1-shot | 5-shot | 1-shot | 5-shot |
| Pixels | 41.7% | 63.2% | 26.7% | 42.6% |
| Baseline Classifier | 80.0% | 95.0% | 69.5% | 89.1% |
| Neural Statistician (Edwards & Storkey, 2016)* | - | - | 88% | 95% |
| Matching Nets (non-FCE, no fine-tune) | **98.1**% | 98.9% | 93.8% | 98.5% |
| Prototypical Nets (1-shot) | **98.1**% | **99.5**% | **94.2**% | **98.6**% |

Table 1: Omniglot few-shot classification accuracy. *Note that the Neural Statistician used non-standard class splits.

### 4.2 *mini*IMAGENET

The *mini*ImageNet dataset (Vinyals et al., 2016) is derived from the larger ImageNet dataset (Deng et al., 2009). It consists of 60,000 color images of size 84 X 84 divided into 100 classes with 600 examples each. It is designed for testing one-shot learning algorithms, where 80 classes are chosen for training, and 20 for testing.

Classification results for *mini*ImageNet are shown in Table 2. The embedding architecture we used for miniImagenet is the same as our experiments for Omniglot, though here it results in a 1600-dimensional output space due to the increased size of the images. We trained two versions of prototypical networks, one with episodes containing a single support examples per class (denoted by 1-shot) and one with five support examples per class (denoted by 5-shot). All episodes contained 20 randomly sampled classes, as 20-way classification represents a more difficult task than 5-way. We evaluated both models on 1-shot and 5-shot for 5-way and 20-way classification at test and find that each model performs best on the number of support examples it was trained for.

| Model | 5-way | | 20-way | |
|---|---|---|---|---|
| | 1-shot | 5-shot | 1-shot | 5-shot |
| Pixels | 23.0% | 26.6% | 6.7% | 7.8% |
| Baseline Classifier | 36.6% | 46.0% | - | - |
| Matching Nets (non-FCE, no fine-tune) | **41.2%** | 56.2% | - | - |
| Prototypical Nets (1-shot) | 40.6% | 55.8% | **16.5%** | 27.5% |
| Prototypical Nets (5-shot) | 37.0% | **57.0%** | 14.2 % | **29.4%** |

Table 2: *mini*ImageNet classification accuracy

### 4.3 CUB ZERO-SHOT CLASSIFICATION

In order to assess the suitability of our approach for zero-shot learning, we also run experiments on the Caltech-UCSD Birds (CUB) 200-2011 dataset (Welinder et al., 2010). In the zero-shot setting, the goal is to classify query images in the absence of any support examples. Instead, class metadata (such as attributes or a textual description) is provided for each of the test classes. We adapt our

| Method | Image Features | Top-1 Acc (50-way) |
|---|---|---|
| ALE (Akata et al., 2013) | Fisher Vectors | 26.9% |
| SJE (Akata et al., 2015) | AlexNet | 40.3% |
| Sample-Clustering (Liao et al., 2016) | AlexNet | 44.3% |
| SJE (Akata et al., 2015) | GoogLeNet | 50.1% |
| DS-SJE (Reed et al., 2016) | GoogLeNet | 50.4% |
| DA-SJE (Reed et al., 2016) | GoogLeNet | 50.9% |
| Prototypical Networks | GoogLeNet | **54.6%** |

Table 3: CUB-200 zero-shot classification accuracy for methods utilizing attribute vectors as class metadata.

few-shot approach to the zero-shot setting by learning to jointly embed images and metadata in a shared space. The embedded metadata serve as class prototypes and classification is performed by embedding the query image and selecting the class whose prototype is nearest in the Euclidean space.

The CUB dataset contains 11,788 images of 200 bird species. We closely follow the procedure of Reed et al. (2016) in preparing the data. We use their splits to divide the classes into disjoint sets of 100 training, 50 validation, and 50 test. For images we use 1,024-dimensional features extracted by applying GoogLeNet (Szegedy et al., 2015) to middle, upper left, upper right, lower left, and lower right crops of the original and horizontally-flipped image[1]. At test time we use only the middle crop of the original image. For class metadata we use the 312-dimensional continuous attribute vectors provided with the CUB dataset. These attributes encode various characteristics of the bird species such as their color, shape, and feather patterns.

We learned a simple linear mapping on top of both the 1,024-dimensional image features and the 312-dimensional attribute vectors to produce a 1,024-dimensional output space. We apply prototype normalization to the embedded attributes so that the class prototypes are always of unit length. This serves as a form of regularization to help our embedding functions generalize better. The model parameters were optimized according to our objective via SGD with Adam (Kingma & Ba, 2014) at learning rate of $10^{-4}$ and weight decay of $10^{-5}$. Early stopping on validation loss was used to determine the optimal number of epochs for retraining on the training + validation set.

Table 3 shows that of methods utilizing attributes as class metadata, we achieve state-of-the-art results by a large margin. Our approach is much simpler than that of other recent approaches (Liao et al., 2016) which train an SVM on a learned feature space obtained by fine-tuning AlexNet (Krizhevsky et al., 2012). These zero-shot classification results demonstrate that our approach is general enough to be applied even when the data points (images) are from a different domain relative to the classes (attributes).

Figure 1 shows a t-SNE (Maaten & Hinton, 2008) visualization of attribute embeddings learned using prototypical networks for zero-shot classification. We can see that the embeddings group the bird species by characteristics such as their color and shape.

## 5 CONCLUSION

We have proposed a simple method called prototypical networks for few-shot learning based on the idea that we can represent each class by the mean of its examples in a representation space learned by a neural network. We train these networks to specifically perform well in the few-shot setting by using episodic training. Prototypical networks are simple to implement, and computationally efficient. We showed that this approach is equivalent to predicting the weights of a linear classifier, where the weights and biases are a function of the prototypes. Prototypical networks achieve state-of-the-art results on the Omniglot dataset, and competitive results on the *mini*Imagenet dataset. We further showed how this approach can be adapted to the zero-shot setting by taking an embedding of an attribute vector for each class to be the prototype. This approach achieves state-of-the-art results on zero-shot classification of the Caltech UCSD birds dataset.

---

[1]https://github.com/reedscot/cvpr2016.

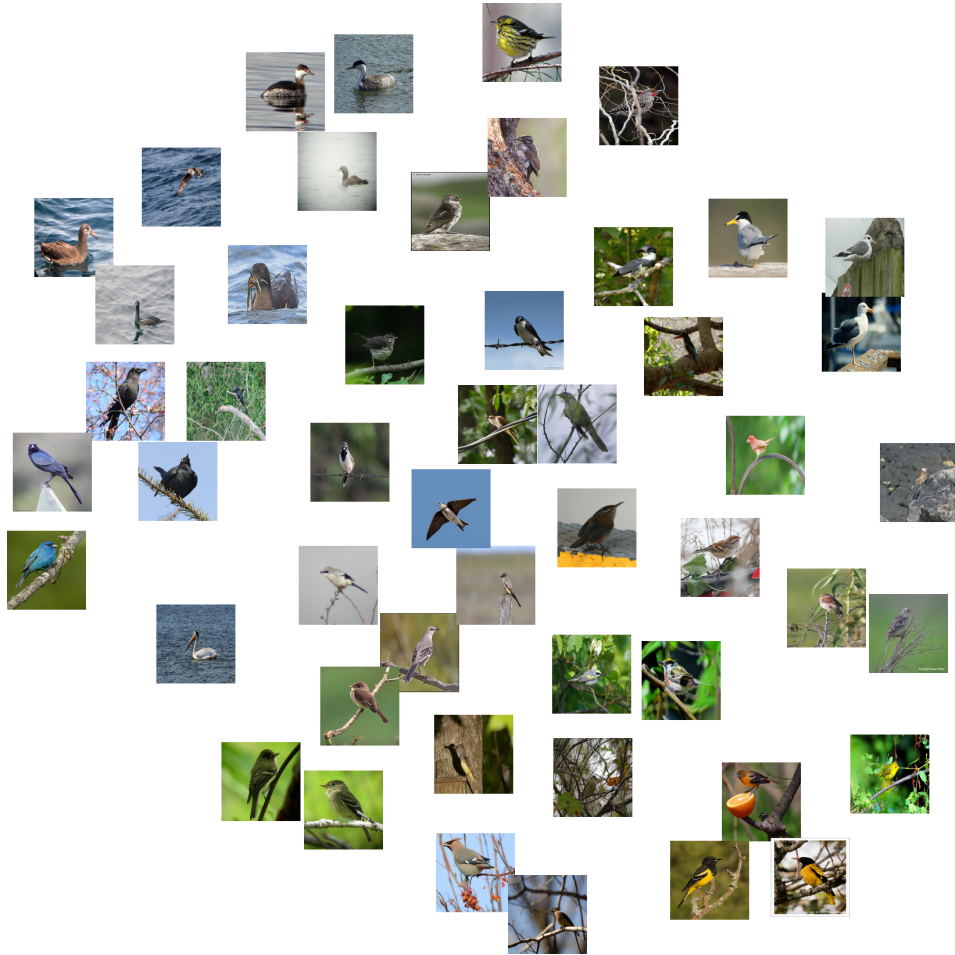

Figure 1: A t-SNE visualization of the attribute embeddings learned by a prototypical network on the CUB dataset. Each image is an arbitrarily chosen example from the corresponding test class. The learned space successfully clusters unseen bird species by characteristics such as color, shape, and pattern.

ACKNOWLEDGMENTS

We would like to thank Sachin Ravi and Hugo Larochelle for help in setting up the Omniglot and *mini*Image data. We would also like to thank Renjie Liao for assistance with the CUB-200 zero-shot procedure and Oriol Vinyals for confirming details regarding the Omniglot and *mini*Imagenet splits and matching nets architectures.

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
