# Peer review of "Prototypical Networks for Few-shot Learning"

_ICLR 2017 — rejected_

[Official Review · AnonReviewer4 · rating 4 · confidence 5 · 15 Dec 2016]
**Overall, paper could greatly be improved, both in the discussion of related work and with a less partial reporting of prior empirical results.**

*** Paper Summary ***

This paper simplify matching network by considering only a single prototype per class which is obtained as the average of the embedding of the training class samples. Empirical comparisons with matching networks are reported.

*** Review ***

The paper reads well and clearly motivate the work. This work of learning metric learning propose to simplify an earlier work (matching network) which is a great objective. However, I am not sure it achieve better results than matching networks. The space of learning embeddings to optimize nearest neighbor classification has been explored before, but the idea of averaging the propotypes is interesting (as a non-linear extension of Mensink et al 2013). I would suggest to improve the discussion of related work and to consolidate the results section to help distinguish between the methods you outperform and the one you do not. 

The related work section can be extended to include work on learning distance metric to optimize a nearest neighbor classification, see Weinberger et al, 2005 and subsequent work. Extensions to perform the same task with neural networks can be found in Min et al, 09 that purse a goal very close to yours. Regarding approaches pursuing similar goals with a different learning objective, you cite siamese network with pairwise supervision. The learning to rank (for websearch) litterature with triplet supervision or global ranking losses is also highly relevant, ie. one example "the query" defines the class and the embedding space need to be such that positive/relevant document are closer to the query than the others. I would suggest to start with Chris Burges 2010 tutorial. One learning class 

I am not sure the reported results correctly reflect the state of the art for all tasks. The results are positive on Omniglot but I feel that you should also report the better results of matching networks on miniImageNet with fine tuning and full contextual embeddings. It can be considered misleading not to report it. On Cub 200, I thought that the state-of-the-art was 50.1%, when using features from GoogLeNet (Akata et al 2015), could you comment on this?

Overall, paper could greatly be improved, both in the discussion of related work and with a less partial reporting of prior empirical results.

*** References ***

Large Margin Nearest Neighbors. Weinberger et al, 2005
From RankNet to LambdaRank to LambdaMART: An Overview, Chris J.C. Burges, June 23, 2010
A Deep Non-linear Feature Mapping for Large-Margin kNN Classification, Min et al, 09

[Official Review · AnonReviewer3 · rating 6 · confidence 4 · 16 Dec 2016]
**Simple but useful extension of matching networks**

This paper proposes an improved version of matching networks, with better scalability properties with respect to the support set of a few-shot classifier. Instead of considering each support point individually, they learn an embedding function that aggregates over items of each class within the support set (eq. 1). This is combined with episodic few-shot training with randomly-sampled partitions of the training set classes, so that the training and testing scenarios match closely.

Although the idea is quite straightforward, and there are a great many prior works on zero-shot and few-shot learning, the proposed technique is novel to my knowledge, and achieves state-of-the-art results on several  benchmark datasets. One addition that I think would improve the paper is a clearer description of the training algorithm (perhaps pseudocode). In its current form the paper a bit vague about this.

[Official Review · AnonReviewer2 · rating 5 · confidence 3 · 19 Dec 2016]
**Extension of matching networks but cannot see big advantages**

The paper is an extension of the matching networks by Vinyals et al. in NIPS2016. Instead of using all the examples in the support set during test, the method represents each class by the mean of its learned embeddings. The training procedure and experimental setting are very similar to the original matching networks. I am not completely sure about its advantages over the original matching networks. It seems to me when dealing with 1-shot case, these two methods are identical since there is only one example seen in this class, so the mean of the embedding is the embedding itself. When dealing with 5-shot case, original matching networks compute the weighted average of all examples, but it is at most 5x cost. The experimental results reported for prototypical nets are only slightly better than matching networks. I  think it is a simple, straightforward,  novel extension, but I am not fully convinced its advantages.

[Final Decision · Program Chairs · 06 Feb 2017]
**ICLR committee final decision**

The program committee appreciates the authors' response to concerns raised in the reviews. Unfortunately, reviews are not leaning sufficiently towards acceptance. Reviewers have concerns about the relationships of this work to existing work in literature (both in terms of a discussion to clarify the novelty, and in terms of more complete empirical comparisons). Authors are strongly encouraged to incorporate reviewer feedback in future iterations of the work.